# The incidence of chronic pain following Cesarean section and associated risk factors: A cohort of women followed up for three months

Natália Carvalho Borges[1]☯*, José Miguel de Deus[2]☯, Rafael Alves Guimarães[1]‡, Délio Marques Conde[2]‡, Maria Márcia Bachion[1]‡, Louise Amália de Moura[1]‡, Lilian Varanda Pereira[1]☯

1 School of Nursing, Federal University of Goiás, Goiânia, Goiás, Brazil, 2 Department of Obstetrics and Gynecology, Federal University of Goiás, Goiânia, Goiás, Brazil

☯ These authors contributed equally to this work.
‡ These authors also contributed equally to this work.
* nataliacb.enf@gmail.com

**Data Availability Statement:** All relevant data are within the manuscript and its Supporting Information files.

## Abstract

### Background

Chronic post-surgical pain (CPSP) is one of the post-surgical complications of a Cesarean section. Despite the high rates of Cesarean section worldwide, the incidence of CPSP and the risk factors for this condition remain relatively unknown. The objective of this study was to calculate the incidence of CPSP in women submitted to Cesarean section and to analyze the associated risk factors.

### Materials and methods

A prospective cohort of 621 women undergoing Cesarean section was recruited preoperatively. Potential presurgical (sociodemographic, clinical and lifestyle-related characteristics) and post-surgical risk factors (the presence and intensity of pain) risk factors were analyzed. Pain was measured at 24 hours and 7, 30, 60 and 90 days after surgery. Following discharge from hospital, data were collected by telephone. The outcome measure was self-reported pain three months after a Cesarean section. The risk factors for chronic pain were analyzed using the log-binomial regression model (a generalized linear model).

### Results

A total of 462 women were successfully contacted 90 days following surgery. The incidence of CPSP was 25.5% (95%CI: 21.8–29.7). Risk factors included presurgical anxiety (adjusted relative risk [RR] 1.03; 95%CI: 1.01–1.05), smoking (adjusted RR 2.22; 95%CI: 1.27–3.88) and severe pain in the early postoperative period (adjusted RR 2.79; 95%CI: 1.29–6.00).

**Funding:** This work was supported by Fundação de Amparo à Pesquisa do Estado de Goiás - FAPEG (www.fapeg.go.gov.br), grant number: 201410267000318 to LVP. NCB received a PhD scholarship from FAPEG, grant number: 88887.162732/2018-00. The funder had no role in study design, data collection and analysis, decision to publish or prepare the manuscript.

**Competing interests:** The authors have declared that no competing interests exist.

## Conclusion

One in four women submitted to Cesarean section may develop CPSP; however, the risk factors identified here are modifiable and preventable. Preventive strategies directed towards controlling anxiety, reducing smoking during pregnancy and managing pain soon after hospital discharge are recommended.

## Introduction

Chronic post-surgical pain (CPSP) is one of the most common and important post-surgical complications [1]. Little is known on the mechanisms underlying the development of this pain [2, 3] and there is no consensus regarding the prevalence rates of CPSP in the different forms of surgery [4], including Cesarean section.

The reported incidence of CPSP following Cesarean section has ranged from less than 1% to 23% one year after surgery [5–10]. Despite the low incidence reported in some studies [9, 10], CPSP could represent a public health issue due to current obstetric practices worldwide that ultimately favor high rates of Cesarean section. In 2015, around 29 million Cesarean sections were performed worldwide, representing an increase of 16 million compared to the figures for the year 2000 [11]. These numbers are concerning, since, although Cesarean sections can reduce maternal and fetal morbidity and mortality [12], they can also trigger unfavorable health outcomes, including CPSP [6, 10, 13–15].

In women submitted to Cesarean section, CPSP may exert a negative effect on activities of daily living [5, 13] and quality of life [7], thus representing an important clinical problem. Consequently, the World Health Organization has included CPSP in the new International Classification of Diseases (ICD-11) [16].

Because CPSP is often difficult to treat [2], identifying the risk factors in different populations is crucial. There appears to be a consensus that acute post-surgical pain is a relevant contributing factor to the development of CPSP in women submitted to Cesarean section [5, 7, 17, 18]. However, the literature on the factors potentially involved in this outcome remains insufficient for any firm conclusions to be reached [19]. The objective of the present study was to calculate the incidence of CPSP in women submitted to Cesarean section and to analyze the risk factors involved.

## Material and methods

### Design and setting

This prospective cohort study was nested within a larger study entitled: "*Evaluation of persistent post-surgical pain in a cohort of women submitted to Cesarean section*". Participants were recruited in a medium-sized private hospital in a city of central Brazil between February 2014 and July 2015. The internal review board of the Federal University of Goiás approved the study protocol under reference number 421,825. The study was conducted in accordance with the international ethical principles for studies involving human participants. All the participating women gave their written informed consent.

### Participants, eligibility criteria and reasons for discontinuation

For the larger main study, 1,122 women who were admitted to hospital for a Cesarean section were recruited. The present study included only those who underwent an elective Cesarean section and who did not meet the following exclusion criteria: age <20 years, presence of pain

in the pelvic region that preceded pregnancy, women who had requested that tubal ligation be performed during the Cesarean section, women who had difficulty in communicating verbally, and those in continuous use of opioids.

Participants who could not be contacted on the date scheduled for the interviews following discharge were classified as lost to follow-up after at least three attempts had been made to contact them. However, participants unable to be contacted previously were permitted to participate in subsequent interviews if later attempts to contact them were successful.

## Data collection

Trained investigators conducted all the interviews and a standardized instrument was used to register the data obtained. Immediately prior to surgery, the data collected consisted of the women's socioeconomic characteristics (age, marital status, schooling, whether in paid employment, and socioeconomic classification), their clinical data (presurgical pain, previous surgeries and childbirths, anxiety and depression) and lifestyle-related data (practice of physical activity, alcohol consumption and smoking). Immediately following surgery (at 24 hours after surgery), data were obtained on variables regarding the occurrence and intensity of postsurgical pain. Subsequently, in the early postoperative period (day 7 following surgery) and at 30, 60 and 90 days after surgery, data on the occurrence and intensity of pain were also collected. After day 7, the interviews were all conducted by telephone. When the women reported no pain at 30 and 60 days following Cesarean section, the outcome variable was automatically considered to be absent on day 90.

## Instruments

The Brazilian Criteria of Economic Classification serves to classify individuals according to their purchasing power [20]. The instrument generates a score that ranges from 0 to 46 according to the number of household appliances and vehicles, type of housing, the presence of a full-time domestic servant and the education level of the head of the family. Individuals are classified as belonging to classes A1, A2, B1, B2, C1, C2, D or E, with class A referring to the highest socioeconomic class and class E to the lowest. For the purposes of this study, the classes were grouped together into A/B, C and D/E.

**Numerical pain scale.** A numerical pain scale was used to measure the intensity of pain, with a scale of points ranging from 0 to 10 in which 0 represents no pain at all, 1–3 represent mild pain, 4–6 moderate pain and 7–10 severe pain [21].

**Hospital Anxiety and Depression Scale (HADS).** The HADS is used to investigate the presence or absence of clinically relevant levels of anxiety and depression. The scale is composed of 14 items, 7 of which refer to the state of anxiety (HADS-A subscale) and 7 to the symptoms of depression (HADS-D subscale). Scores, which can range from 0 to 21 points, reflect the individual's current mood, particularly over the preceding week [22]. In both subscales, scores ≥9 are suggestive of anxiety and/or depression. A version of HADS translated into Brazilian Portuguese and adapted for use in the country was used in this study [23]. The internal consistency of this version has been found to be good and levels of sensitivity and specificity are high [24].

**State-Trait Anxiety Inventory (STAI).** Only the trait anxiety subscale (STAI-T) of this instrument, which serves to evaluate trait anxiety (anxious personality), was used in this study. This instrument consists of 20 items, and the overall score ranges from 20 to 80 points, with higher scores being indicative of greater levels of anxiety [25]. The version used in the present study was previously translated and validated for use in Brazilian Portuguese and its reliability has been reported to be good [26, 27].

## Statistical analysis

In a descriptive analysis, the quantitative variables were presented as means and standard deviations (SD) and the qualitative variables as absolute and relative frequencies. The incidence of CPSP was calculated from the number of women who developed chronic pain ≥3 months following a Cesarean section, divided by the total number of women x 100. In the analysis of the possible risk factors for CPSP, missing values for the variables of interest were calculated using multiple imputation techniques in order to reduce analytical biases [28]. Data were imputed for the following variables: schooling (0.4%), pregnancy-associated presurgical pain (0.2%), anxiety (STAI-T) (1.7%), anxiety (HADS-A) (2.4%), depression (HADS-D) (2.4%), duration of surgery (0.4%), pain at 24 hours following surgery (0.4%) and pain at 7 days following surgery (18.2%). A log-binomial regression model (a generalized linear model) was used to analyze the factors associated with the risk of chronic pain [29, 30]. Sociodemographic, clinical and lifestyle-related characteristics collected prior to surgery and the variables associated with pain at 24 hours and 7 days following surgery were defined as independent variables. Initially, a bivariate analysis was performed to investigate the association between the study outcome measure and each independent variable analyzed. Next, variables with p-values <0.20 were included in a multivariate log-binomial regression model with robust variance estimation. Possible interactions between the variables were tested for significance. Variables were allowed to remain in the model if statistically significant (p<0.05). The results of the final regression model were presented as adjusted relative risks (RR) and their respective confidence intervals (95%CI).

## Results

Based on the eligibility criteria, 648/1,122 women were selected for the present study; however, 27 refused to participate, leaving a total of 621 participants. Mean age at baseline was 26.4 ± 4.8 years. Most (89.2%) were in a stable relationship and 64.9% had been submitted to a Cesarean section for the first time (Table 1). The reasons for discontinuation from the study were classified as: death of the newborn infant (n = 2), dropout from the study (n = 4) and reported symptoms of infection at the site of surgery (n = 47) (Fig 1).

Anesthesia in all cases consisted of intrathecal bupivacaine at a mean dose of 12.3 ± 1.3 mg together with morphine at a mean dose of 87.2 ± 11.9 mcg. In some cases (n = 19; 3.1%), fentanyl was added to this intrathecal combination at a mean dose of 21.5 mcg. Approximately half the participants (53.1%) received only intrathecal analgesia during surgery, while the remainder (46.9%) received additional analgesia administered intravenously. Of the analgesic agents used, dipyrone was administered in 89.9% of cases at a mean dose of 1.6 ± 0.4 g. The surgical technique of choice was the Pfannenstiel incision in all cases and the mean duration of surgery was 35 ± 11 minutes.

Of the 620 women monitored after surgery, all received some type of medication for pain relief while in hospital. As shown in S1 Table, almost all (99.7%) were given simple analgesics and non-steroidal anti-inflammatory drugs (NSAIDs) (93.3%) during this period. Of the simple analgesics, dipyrone was the most commonly administered medication (99.2%), while, of the NSAIDs, diclofenac sodium was the most common (99.5%). Only five women (0.8%) were given an opioid analgesic during this period.

Following discharge from hospital, 95.8%, 36.0%, 15.5% and 17.0% of the women with surgery-related pain used some form of pain relief medication on the 7th, 30th, 60th and 90th day after the Cesarean section, respectively. Of the women who used simple analgesics, dipyrone was the drug of choice at all the evaluation moments (7th day = 95.9%; 30th day = 87.2%; 60th day = 91.7% and 90th day = 90.9%). Of those who used an NSAID, diclofenac sodium was the

**Table 1. Sociodemographic, clinical and lifestyle characteristics of the study participants (n = 621).**

| Variables | | |
|---|---|---|
| Age (years), mean (SD) | 26.4 | (4.8) |
| Marital status, n (%) | | |
| Has a stable partner | 554 | (89.2) |
| Does not have a stable partner | 67 | (10.8) |
| Schooling [a], n (%) | | |
| $\geq$ 8 years | 573 | (92.6) |
| < 8 years | 46 | (7.4) |
| In paid employment, n (%) | | |
| Yes | 406 | (65.4) |
| No | 215 | (34.6) |
| Socioeconomic class [b], n (%) | | |
| AB | 210 | (33.9) |
| C | 368 | (59.3) |
| DE | 42 | (6.8) |
| Report of chronic pain prior to surgery [b], n (%) | | |
| Yes | 375 | (60.5) |
| No | 245 | (39.5) |
| Report of pregnancy-related presurgical pain [c], n (%) | | |
| Yes | 334 | (54.0) |
| No | 284 | (46.0) |
| Report of pain at the time of the presurgical interview [b], n (%) | | |
| Yes | 162 | (26.1) |
| No | 458 | (73.9) |
| Previous Cesarean section, n (%) | | |
| Yes | 218 | (35.1) |
| No | 403 | (64.9) |
| Previous vaginal delivery, n (%) | | |
| Yes | 104 | (16.7) |
| No | 517 | (83.3) |
| Other previous surgeries [d], n (%) | | |
| Yes | 37 | (6.0) |
| No | 579 | (94.0) |
| Anxiety [e] (STAI-T), mean (SD) | 39.5 | (8.6) |
| Anxiety [f] (HADS-A), mean (SD) | 7.6 | (4.0) |
| Anxiety [f] (HADS-A), n (%) | | |
| Yes ($\geq$ 9) | 237 | (39.0) |
| No (< 9) | 371 | (61.0) |
| Depression [f] (HADS-D), mean (SD) | 4.8 | (3.2) |
| Depression [f] (HADS-D), n (%) | | |
| Yes ($\geq$ 9) | 79 | (13.0) |
| No (< 9) | 529 | (87.0) |
| Practice of physical exercise, n (%) | | |
| Yes | 47 | (7.6) |
| No | 574 | (92.4) |
| Alcohol consumption, n (%) | | |
| Yes | 36 | (5.8) |
| No | 585 | (94.2) |

(*Continued*)

**Table 1.** (Continued)

| Variables | | |
|---|---|---|
| Smoking, n (%) | | |
| Yes | 14 | (2.3) |
| No | 607 | (97.7) |

[a] Data missing = 2

[b] Data missing = 1

[c] Data missing = 3

[d] Data missing = 5

[e] Data missing = 9

[f] Data missing = 13.

most commonly used drug throughout follow-up (7th day = 97.9%; 30th day = 92.3%; 60th day = 90.0% and 90th day = 71.4%). None of the participants used an opioid following discharge from hospital (S2–S5 Tables).

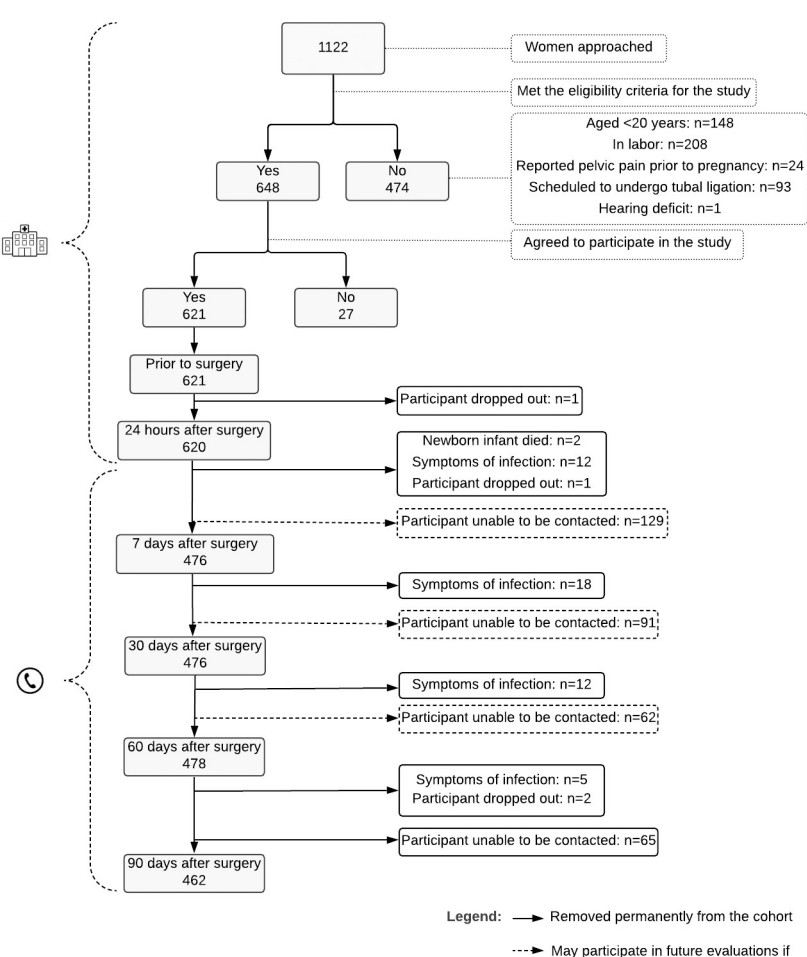

**Fig 1. Flowchart of recruitment to the study.**

Three months after Cesarean section, the incidence of CPSP was 25.5% (95%CI: 21.8–29.7). When the women with CPSP (n = 118) were asked about the intensity of pain at the moment when this symptom was most intense, 16.1%, 47.5% and 36.4% reported mild, moderate and severe pain, respectively. The mean intensity of pain was 5.7 (±2.3 SD).

In the bivariate analysis, age, years of schooling, smoking, anxiety (STAI-T), depression and severe pain 7 days following surgery were associated with the incidence of CPSP (Table 2).

Table 3 shows the final multivariate log-binomial regression model for the risk factors for CPSP. The presence of chronic pain in this cohort was significantly associated with smoking (adjusted RR: 2.22; 95%CI: 1.27–3.88), preoperative anxiety (STAI-T) (adjusted RR: 1.03; 95% CI: 1.00–1.05) and severe post-surgical pain 7 days after surgery (adjusted RR: 2.79; 95%CI: 1.29–6.00).

## Discussion

In the present study, the incidence of CPSP in women submitted to Cesarean section was considered high, affecting one in four participants after three months of follow-up. In view of the harm to women caused by this painful condition and the difficulty in obtaining effective treatment, efforts should be aimed particularly at identifying women at a greater risk of developing CPSP. The results of this study add to the already available knowledge on the subject, showing that a greater intensity of pain seven days after surgery, smoking, and the presence of anxiety prior to surgery can all be considered factors that increase the risk of women presenting with this complication.

The high incidence found in the present study reflects a relevant public health issue due to the current situation in obstetrics worldwide in which an unprecedented increase has been seen in the frequency with which Cesarean sections are performed [31]. The 25.5% incidence of CPSP found in this study is similar to that reported from another three prospective studies conducted to evaluate pain three months following Cesarean section, with rates in those studies ranging from 22.5 to 30.7% [7, 14, 32].

In another similar study, an incidence as low as 6.8% was reported [10]. Those authors argued that by having excluded women who had reported pain prior to surgery they had avoided a possible overestimation of the incidence due to the continuation of preexisting pain. In the present study, however, to avoid this type of bias, only women with pain in the pelvic region prior to pregnancy were excluded. Women who had reported pain in other regions of the body were allowed to remain in the study, since that pain could represent a risk factor for CPSP [32–34]. Furthermore, due to the physiological and structural changes that occur during pregnancy, pain, particularly lower back pain, is often reported [35, 36]. Therefore, by excluding all participants with presurgical pain, the incidence reported would not reflect the clinical reality.

In contrast with the findings of other studies, presurgical pain and pain at 24 hours following surgery were not associated with CPSP in the present study [5, 6, 8, 9, 32–34, 37]. On the other hand, severe pain seven days following surgery was found to represent a risk factor for CPSP. To the best of our knowledge, no other studies have investigated this association, an important finding that raises questions regarding the management of pain in new mothers following discharge from hospital. The present findings are in line with current knowledge regarding the possible physiological mechanisms of the chronification of post-surgical pain in which persistent nociceptive signaling from peripheral tissue is able to modulate pain, leading to the chronification of pain [3].

Even apart from Cesarean sections, few studies have investigated early postoperative pain in other types of surgery, and this is an area that has received scant attention despite

**Table 2. Bivariate analysis of the potential risk factors for chronic post-surgical pain in women submitted to Cesarean section.**

| | | Total (n = 462) | Incidence | | Unadjusted RR | 95%CI | p-value |
|---|---|---|---|---|---|---|---|
| | | | n = 118 | x100 women | | | |
| **Sociodemographic factors** | | | | | | | |
| Age (years) (mean ± SD) CV | | 26.3 (4.8) | 27.1 (5.4) | | 1.03 | 1.01–1.07 | 0.040 |
| Has a stable partner | Yes | 414 | 107 | 25.8 | 1.00 | | |
| | No | 48 | 11 | 22.9 | 0.89 | 0.51–1.53 | 0.665 |
| Schooling ≥ 8 years | Yes | 428 | 104 | 24.3 | 1.00 | | |
| | No | 34 | 14 | 41.2 | 1.70 | 1.10–2.62 | 0.018 |
| In paid employment | Yes | 302 | 84 | 27.8 | 1.00 | | |
| | No | 160 | 34 | 21.3 | 0.74 | 0.54–1.08 | 0.131 |
| Socioeconomic class | AB | 154 | 33 | 21.4 | 1.00 | | |
| | C | 272 | 74 | 27.2 | 1.27 | 0.89–1.82 | 0.193 |
| | DE | 36 | 11 | 30.6 | 1.43 | 0.80–2.54 | 0.229 |
| **Clinical factors** | | | | | | | |
| Report of chronic pain prior to surgery | Yes | 280 | 68 | 24.3 | 0.88 | 0.65–1.21 | 0.441 |
| | No | 182 | 50 | 27.5 | 1.00 | | |
| Pregnancy-related presurgical pain | Yes | 237 | 67 | 28.3 | 1.25 | 0.91–1.71 | 0.170 |
| | No | 225 | 51 | 22.7 | 1.00 | | |
| Reported pain at the presurgical interview | None | 349 | 87 | 24.9 | 1.00 | | |
| | Mild | 5 | 2 | 40.0 | 1.60 | 0.54–4.77 | 0.395 |
| | Moderate | 35 | 8 | 22.9 | 0.92 | 0.49–1.73 | 0.789 |
| | Severe | 73 | 21 | 28.8 | 1.15 | 0.77–1.73 | 0.487 |
| Previous Cesarean section | Yes | 158 | 36 | 22.8 | 0.85 | 0.60–1.19 | 0.333 |
| | No | 304 | 82 | 27.0 | 1.00 | | |
| Previous vaginal delivery | Yes | 74 | 16 | 21.6 | 0.82 | 0.52–1.31 | 0.410 |
| | No | 388 | 102 | 26.3 | 1.00 | | |
| Other previous surgeries | Yes | 173 | 37 | 21.4 | 0.76 | 0.54–1.07 | 0.119 |
| | No | 289 | 81 | 28.0 | 1.00 | | |
| Anxiety (STAI-T) (mean ± SD) | | 39.1 (8.6) | 41.7 (9.6) | | 1.03 | 1.02–1.05 | < 0.001 |
| Anxiety (HADS-A), mean ± SD | | 7.5 (3.9) | 8.0 (4.1) | | 1.04 | 0.99–1.08 | 0.067 |
| Depression (HADS-D), mean ± SD | | 4.6 (3.0) | 5.1 (3.1) | | 1.05 | 1.00–1.10 | 0.042 |
| Duration of surgery (minutes), mean ± SD | | 35.4 (11.0) | 34.8 (10.5) | | 0.99 | 0.98–1.01 | 0.539 |
| Pain 24 hours after surgery | None | 37 | 7 | 18.9 | 1.00 | | |
| | Mild | 35 | 6 | 17.1 | 0.91 | 0.34–2.43 | 0.845 |
| | Moderate | 168 | 40 | 23.8 | 1.26 | 0.61–2.58 | 0.531 |
| | Severe | 222 | 65 | 29.3 | 1.55 | 0.77–3.11 | 0.220 |
| Pain at 7 days after surgery | None | 54 | 6 | 11.1 | 1.00 | | |
| | Mild | 44 | 8 | 18.2 | 1.64 | 0.61–4.36 | 0.325 |
| | Moderate | 176 | 36 | 20.5 | 1.84 | 0.82–4.13 | 0.139 |
| | Severe | 188 | 68 | 36.2 | 3.35 | 1.49–7.10 | 0.003 |
| **Lifestyle factors** | | | | | | | |
| Practice of physical activity | Yes | 31 | 7 | 22.6 | 1.00 | | |
| | No | 431 | 111 | 25.8 | 1.14 | 0.58–2.23 | 0.701 |
| Alcohol consumption | Yes | 24 | 5 | 20.8 | 0.81 | 0.36–1.79 | 0.599 |
| | No | 438 | 113 | 25.8 | 1.00 | | |
| Smoking | Yes | 5 | 3 | 60.0 | 2.38 | 1.15–4.96 | 0.020 |
| | No | 457 | 115 | 25.2 | 1.00 | | |

RR: unadjusted relative risk; 95%CI: 95% confidence interval; CV = Continuous variable.

**Table 3. Multivariate regression model for the risk factors for chronic post-surgical pain in women submitted to Cesarean section.**

| | | Adjusted RR* | 95%CI | ß | p-value |
|---|---|---|---|---|---|
| Age (years) | | 1.03 | 0.99–1.06 | 0.03 | 0.129 |
| Schooling (years) | ≥ 8 | 1.00 | | | |
| | < 8 | 1.05 | 0.64–1.71 | 0.04 | 0.859 |
| In paid employment | No | 1.00 | | | |
| | Yes | 0.71 | 0.49–1.02 | -0.35 | 0.063 |
| Socioeconomic class | AB | 1.00 | | | |
| | C | 1.26 | 0.88–1.80 | 0.23 | 0.211 |
| | DE | 1.39 | 0.75–2.57 | 0.33 | 0.290 |
| Reported pain at the time of interview | No | 1.00 | | | |
| Anxiety (STAI-T) | CV | 1.03 | 1.01–1.05 | 0.03 | **0.006** |
| Anxiety (HADS-A) | CV | 0.98 | 0.94–1.03 | -0.02 | 0.455 |
| Depression (HADS-D) | CV | 1.00 | 0.94–1.07 | 0.00 | 0.898 |
| Pain at 7 days after surgery | None | 1.00 | | | |
| | Mild | 1.33 | 0.52–3.41 | 0.28 | 0.558 |
| | Moderate | 1.65 | 0.74–3.65 | 0.50 | 0.218 |
| | Severe | 2.79 | 1.29–6.00 | 1.02 | **0.009** |
| | Yes | 1.29 | 0.95–1.77 | 0.26 | 0.105 |
| Smoking | No | 1.00 | | | |
| | Yes | 2.22 | 1.27–3.88 | 0.80 | **0.005** |

RR: relative risk; 95%CI: 95% confidence interval; β: Regression coefficient

* Log-binomial regression model adjusted for age, schooling, whether in paid employment, socioeconomic class, reported pain at interview, anxiety (STAI-T), anxiety (HADS-A), depression (HADS-D), pain 7 days after surgery, and smoking. CV = Continuous variables; Parameters of the model: AIC: 588.34; BIC: 646.24; Pearson goodness-of-fit: $\chi^2$: 271.91; p-value = 0.607.

representing a potential source of important information on the process of the chronification of pain [3]. A study conducted in the United States found that the frequency of surgical patients reporting moderate, severe or extreme pain after discharge from hospital is higher than that recorded while the patients are in hospital [38]. The additional challenge regarding how to appropriately manage pain following Cesarean section is indisputable in view of the potential exposure of the newborn infant via breastfeeding to analgesics administered to the mother and the need for the mother to be able to care for the infant [18, 39]. Nevertheless, the adequate treatment of post-surgical pain, particularly aimed at avoiding the occurrence of severe pain, should continue following discharge from hospital in order to minimize the risk of CPSP.

In addition to the intensity of early post-surgical pain, higher levels of anxiety and smoking also appeared as risk factors for CPSP three months after Cesarean section. These findings are noteworthy because of the common co-occurrence of anxiety and smoking [40]. Smoking has received scant attention from investigators conducting research into CPSP. Nevertheless, as also shown in the present study, prospective observational studies have identified smoking as a risk factor for CPSP following a variety of different types of surgery [41–43]. Therefore, based on evidence from clinical, epidemiological and laboratory-based studies, a reciprocal model has been proposed in which the interrelationships between pain and smoking are bidirectional [44, 45]. Nonetheless, few women in the present study smoked and this may have contributed to overestimation of the magnitude of the effect found here. Future studies with appropriate designs aimed at producing more robust evidence should be conducted to fill this gap in current knowledge.

Likewise, anxiety has received little attention in studies conducted to analyze the incidence of chronic pain following Cesarean section. In view of the particular characteristics of this type of surgery and the greater prevalence of certain anxiety disorders during pregnancy [46, 47], it is vital to evaluate the effect of this risk factor for CPSP. Richez et al. [32] reported that presurgical anxiety acted as a risk factor for CPSP with neuropathic characteristics. Those investigators failed to identify any association between anxiety and the overall incidence of CPSP.

Possible cultural and socioeconomic characteristics of the sample, in addition to the study design, could have contributed to the differences between the present results and those reported by Richez et al. [32]. Particular consideration should be given to the use of different instruments to evaluate anxiety. Richez et al. used the HADS-A, thus obtaining anxiety scores relevant to a recent period of time, specifically the preceding week [22]. Nonetheless, a presurgical increase in anxiety levels is expected in women scheduled to undergo elective Cesarean section [48]. In addition to HADS-A, the present study also used STAI-T, which evaluates a relatively stable personality trait that would lead an individual to respond to stress with anxiety [26]. Therefore, taking the objectives of this particular study into consideration, this would appear to be a more appropriate scale, since it better reflects differences in relation to the woman's tendency to respond to potential stressors.

Anxiety indeed appears to be associated to some degree with the occurrence of CPSP following Cesarean section. This effect may occur due to the influence of cognitive and emotional processes in the descending pain modulation in response to noxious stimuli [49]. Neuroimaging studies have shown changes in the emotional processing of anxious and depressed patients, with emotions and pain then going on to share the same regions of the brain [50, 51]. These changes may persist even after the individual has recovered from the depression or anxiety, negatively affecting the experience of pain.

The present study represents an advance in what is known with respect to CPSP in obstetrics; however, there are some limitations associated with the study. One refers to the use of a non-randomized sample, which could negatively affect the generalization of the results to some degree. Furthermore, not having followed up the women who reported having no pain at 30 and 60 days after surgery could have led to an underestimation of the incidence, since CPSP may involve asymptomatic periods [52]. Despite these limitations, one of the advantages of this study is its prospective design, thus minimizing the risk of the memory bias that can occur when patients need to remember pain experienced in the past. In addition, the results provide evidence that severe pain in the early postoperative period represents a risk factor for CPSP, and the exclusion of women scheduled to undergo tubal ligation and those reporting pelvic pain prior to pregnancy may have contributed towards reaching a more precise estimate of the incidence of this complication.

In conclusion, the findings of the present study provide important data on CPSP in women submitted to Cesarean section. One in four women report pain related to surgery three months after the operation, with this complaint being affected by anxiety levels, smoking and by the presence of severe pain in the early postoperative period. The risk factors identified here are modifiable, indicating that implementing preventive strategies could be beneficial. These strategies include providing emotional care and discouraging smoking in the period preceding a Cesarean section and offering better pain control following discharge from hospital for women who have undergone this surgical intervention.

## Supporting information

**S1 Table. Medication received postoperatively during hospitalization.**
(PDF)

**S2 Table. Use of medication by the women with pain on the 7<sup>th</sup> day following surgery (n = 434).**
(PDF)

**S3 Table. Use of medication by the women with pain on the 30<sup>th</sup> day following surgery (n = 173).**
(PDF)

**S4 Table. Use of medication by the women with pain on the 60<sup>th</sup> day following surgery (n = 147).**
(PDF)

**S5 Table. Use of medication by the women with pain on the 90<sup>th</sup> day following surgery (n = 118).**
(PDF)

**S1 File Database and variable dictionary.**
(XLSX)

## Acknowledgments

The authors thank all the women who participated in this study.

## Author Contributions

**Conceptualization:** Natália Carvalho Borges, José Miguel de Deus, Lilian Varanda Pereira.

**Data curation:** Natália Carvalho Borges, José Miguel de Deus, Rafael Alves Guimarães, Maria Márcia Bachion, Louise Amália de Moura, Lilian Varanda Pereira.

**Formal analysis:** Natália Carvalho Borges, José Miguel de Deus, Rafael Alves Guimarães, Lilian Varanda Pereira.

**Funding acquisition:** Lilian Varanda Pereira.

**Investigation:** Natália Carvalho Borges, Lilian Varanda Pereira.

**Methodology:** Natália Carvalho Borges, José Miguel de Deus, Rafael Alves Guimarães, Délio Marques Conde, Maria Márcia Bachion, Lilian Varanda Pereira.

**Project administration:** Natália Carvalho Borges, José Miguel de Deus, Lilian Varanda Pereira.

**Resources:** Natália Carvalho Borges, José Miguel de Deus, Délio Marques Conde, Maria Márcia Bachion, Lilian Varanda Pereira.

**Software:** Natália Carvalho Borges, Rafael Alves Guimarães, Louise Amália de Moura, Lilian Varanda Pereira.

**Supervision:** Natália Carvalho Borges, Lilian Varanda Pereira.

**Validation:** Rafael Alves Guimarães, Délio Marques Conde, Maria Márcia Bachion.

**Visualization:** Natália Carvalho Borges, José Miguel de Deus, Rafael Alves Guimarães, Maria Márcia Bachion, Louise Amália de Moura, Lilian Varanda Pereira.

**Writing – original draft:** Natália Carvalho Borges, Rafael Alves Guimarães, Délio Marques Conde, Lilian Varanda Pereira.

**Writing – review & editing:** Natália Carvalho Borges, José Miguel de Deus, Rafael Alves Guimarães, Délio Marques Conde, Maria Márcia Bachion, Louise Amália de Moura, Lilian Varanda Pereira.

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
