## [Decision Letter · Decision Letter 0]

17 Jun 2020

PONE-D-20-14430

The incidence of chronic pain following Cesarean section and associated risk factors: a cohort of women followed up for three months

PLOS ONE

Dear Dr. 

 To: Natalia Carvalho Borges 

Thank you for submitting your manuscript to PLOS ONE. After careful consideration, we feel that it has merit but does not fully meet PLOS ONE’s publication criteria as it currently stands. Therefore, we invite you to submit a revised version of the manuscript that addresses the points raised during the review process.

I would appreciate if you respond carefully to the reviewers' comments in your reply. 

We look forward to receiving your revised manuscript.

Kind regards,

Ehab Farag, MD FRCA FASA

Academic Editor

PLOS ONE

Journal Requirements:

Additional Editor Comments (if provided):

I would appreciate if you respond carefully to the reviewers' comments in your reply.

Reviewers' comments:

Reviewer's Responses to Questions

**Comments to the Author**

1. Is the manuscript technically sound, and do the data support the conclusions?

Reviewer #1: Yes

Reviewer #2: Yes

2. Has the statistical analysis been performed appropriately and rigorously? 

Reviewer #1: Yes

Reviewer #2: Yes

3. Have the authors made all data underlying the findings in their manuscript fully available?

Reviewer #1: Yes

Reviewer #2: Yes

4. Is the manuscript presented in an intelligible fashion and written in standard English?

Reviewer #1: Yes

Reviewer #2: Yes

5. Review Comments to the Author

Reviewer #1: Line 81 please add "elective" before Cesarean

Is there a role of prior C Section on chronic pain? Please elaborate on the incidence of chronic pain in primary versus repeat C section.

Manuscript well written and comprehensive

Reviewer #2: I congratulate the authors in doing this prospective study. This topic is very important and well described.

1. What analgesics did patients receive in post operative period other than dipyrone?

2. Do you allow partner's presence during cesarean section to reduce anxiety?

3. Do patients receive analgesic when they were in pain after 7 days of surgery?

4. Can you provide the severity of pain at 90 days?

5. Did patients demand analgesic at 60 and 90 days?

6. Any H/O drug abuse in the past for any of the patients?

7. Can you include the surgeon variable? (eg- is there one or two particular surgeons whose patients had less pain or more CPSP?

6. PLOS authors have the option to publish the peer review history of their article (what does this mean?). If published, this will include your full peer review and any attached files.

Reviewer #1: Yes: SABRY AYAD MD MBA FASA

Reviewer #2: Yes: JAGAN DEVARAJAN

---

## [Author Response · Author response to Decision Letter 0]

30 Jul 2020

Reviewer 1

Comment: Line 81 please add "elective" before Cesarean

The reviewer’s suggestion implies that the description of the study population was not clear; therefore, we decided that the text should be improved. The present study sample described in the manuscript was indeed made up of women who had been submitted to an elective Cesarean section, i.e. prior to the onset of labor and rupture of the membranes [1,2]. Nevertheless, the main study in which this study was nested included all the women admitted for a Cesarean section (n=1,122) irrespective of whether or not the procedure was elective. For this reason, we decided not to include the term “elective” before “Cesarean section” in that specific segment, as suggested by the reviewer, since this would lead to the interpretation that only women submitted to an elective Cesarean section had participated in the larger main study, which was not the case. Therefore, to improve our description of the study population, we have rewritten the relevant segment of the Methods section (page 4, lines 80-85), as follows:

“For the larger main study, 1,122 women who were admitted to hospital for a Cesarean section were recruited. The present study included only those who underwent an elective Cesarean section and who did not meet the following exclusion criteria: age <20 years, presence of pain in the pelvic region that preceded pregnancy, women who had requested that tubal ligation be performed during the Cesarean section, women who had difficulty in communicating verbally, and those in continuous use of opioids.”

Comment: Is there a role of prior C Section on chronic pain? Please elaborate on the incidence of chronic pain in primary versus repeat C section

There is some evidence that previous surgeries may play a role in the chronification of postoperative pain [3–6]. This may occur due to repeated exposure to postoperative pain treated with opioids, resulting in central sensitization and hyperalgesia [5] and/or to the increased risk of nerve entrapment caused by the increase in fibroses at repeat surgeries [6]. Consequently, we analyzed the effect of exposure to a prior Cesarean section and the occurrence of chronic postoperative pain. These results are shown in Table 2 (page 12). In agreement with other studies with similar populations [7–13], there was no statistically significant difference between the incidence of chronic postoperative pain in women who had been submitted to a previous C-section and those who were undergoing the procedure for the first time. 

Reviewer 2

Comment: What analgesics did patients receive in postoperative period other than dipyrone?

Thank you for this question. Data regarding the medication received for pain relief were collected postoperatively for all the women in the cohort (n=620) while they were still in hospital. We have now added further information on this subject in the Results section (page 10, lines 179-184), as follows, and have added an extra table (S1 Table) in the Supporting Information section.

“Of the 620 women monitored after surgery, all received some type of medication for pain relief while in hospital. As shown in S1 Table, almost all (99.7%) were given simple analgesics and non-steroidal anti-inflammatory drugs (NSAIDs) (93.3%) during this period. Of the simple analgesics, dipyrone was the most commonly administered medication (99.2%), while, of the NSAIDs, diclofenac sodium was the most common (99.5%). Only five women (0.8%) were given an opioid analgesic during this period”.

Comment: Do you allow partner's presence during cesarean section to reduce anxiety?

Not all of the women were able to have an accompanying person in the operating room, since permission for a companion to be present in the hospital where data collection was performed was dependent on authorization from the attending surgeon. In this study, data are not available on this factor since this information was not collected in a standardized manner.

Comment: Do patients receive analgesic when they were in pain after 7 days of surgery?

Comment: Did patients demand analgesic at 60 and 90 days?

Yes, some of the women were in use of pain relief medication during follow-up. The reviewer’s question made us reconsider how this information was provided in the manuscript and we decided to add a paragraph in the Results section and include some supplementary tables (S2-S5 Tables) regarding the use of pain medication during follow-up.

Since information was included on the use of medication at all evaluation moments during follow-up, we deemed it relevant to also present related data in the flowchart of the cohort. Therefore, we have changed the flowchart to add the number of women evaluated at 30 and 60 days following surgery (Fig 1).

The following paragraph was included in the manuscript (page 10, lines 185-192) and refers to the additional tables:

“Following discharge from hospital, 95.8%, 36.0%, 15.5% and 17.0% of the women with surgery-related pain used some form of pain relief medication on the 7th, 30th, 60th and 90th day after the Cesarean section, respectively. Of the women who used simple analgesics, dipyrone was the drug of choice at all the evaluation moments (7th day = 95.9%; 30th day = 87.2%; 60th day = 91.7% and 90th day = 90.9%). Of those who used an NSAID, diclofenac sodium was the most commonly used drug throughout follow-up (7th day = 97.9%; 30th day = 92.3%; 60th day = 90.0% and 90th day = 71.4%). None of the participants used an opioid following discharge from hospital (S2-S5 Tables)”.

Comment: Can you provide these severity of pain at 90 days?

Yes, the women reporting pain on the 90th day following surgery were asked about the intensity of this symptom, particularly when the pain was most intense. We have added the following segment to the Results section (page 10, lines 194-196).

“When the women with CPSP (n=118) were asked about the intensity of pain at the moment when this symptom was most intense, 16.1%, 47.5% and 36.4% reported mild, moderate and severe pain, respectively. The mean intensity of pain was 5.7 (±2.3 SD)”.

Comment: Any H/O drug abuse in the past for any of the patients?

In relation to drug abuse, prior to surgery the patients were asked about their use of opioids, alcohol and tobacco. All stated that they did not use opioids (an exclusion criterion). Data on alcohol consumption and smoking are shown in the tables. Regarding opioid use, we have now included the following phrase in the manuscript as part of the exclusion criteria in the Methods section (page 4, line 85):

“and those in continuous use of opioids.”.

Comment: Can you include the surgeon variable? (e.g. is there one or two particular surgeons whose patients had less pain or more CPSP?

At this institute, six different surgeons perform Cesarean sections. In the present study, it was not possible to include data on the variable “surgeon” since this information was not collected in a standardized manner.

REFERENCES

1. World Health Organization. (‎2018)‎. WHO recommendation: elective C-sections hould not be routinely recommended to women living with HIV: policy brief. World Health Organization. https://apps.who.int/iris/handle/10665/272454.

2. International Perinatal HIV Group. The mode of delivery and the risk of vertical transmission of human immunodeficiency virus type 1 – a meta-analysis of 15 prospective cohort studies. N Engl J Med. 1999;340:977–987. doi:10.1056/NEJM199904013401301

3. Brandsborg B, Nikolajsen L, Hansen CT, Kehlet H, Jensen TS. Risk factors for chronic pain after hysterectomy: A Nationwide questionnaire and database study. Anesthesiology. 2007;106:1003–1012. doi: 10.1097/01.anes.0000265161.39932.e8

4. Rivat C, Laboureyras E, Laulin J-P, Le Roy C, Richebé P, Simonnet G. Non-nociceptive environmental stress induces hyperalgesia, not Analgesia, in pain and opioid-experienced rats. Neuropsychopharmacology. 2007;32:2217–28. doi:10.1038/sj.npp.1301340

5. Rivat C, Ballantyne J. The dark side of opioids in pain management: basic science explains clinical observation. Pain Reports. 2016;1:e570. doi: 10.1097/PR9.0000000000000570

6. Loos MJ, Scheltinga MR, Mulders LG, Roumen RM. The Pfannenstiel Incision as a Source of Chronic Pain. Obstet Gynecol. 2008;111:839–846. doi:10.1097/AOG.0b013e31816a4efa

7. Nikolajsen L, Sorensen HC, Jensen TS, Kehlet H. Chronic pain following Caesarean section. Acta Anaesthesiol Scand. 2004;48: 111–116. doi:10.1111/j.1399-6576.2004.00271.x

8. Sng BL, Sia ATH, Quek K, Woo D, Lim Y. Incidence and Risk Factors for Chronic Pain after Caesarean Section under Spinal Anaesthesia. Anaesth Intensive Care. 2009;37: 748–52. doi:10.1177/0310057X0903700513

9. Kainu JP, Halmesmaki E, Korttila KT, Sarvela PJ. Persistent pain after cesarean delivery and vaginal delivery: A prospective cohort study. Anesth Analg. 2016;123: 1535–1545. doi:10.1213/ANE.0000000000001619

10. Liu TT, Raju A, Boesel T, Cyna AM, Tan SGM. Chronic pain after caesarean delivery: An Australian cohort. Anaesth Intensive Care. 2013;41: 496–500. doi:10.1177/0310057x1304100410

11. Nardi N, Campillo-Gimenez B, Pong S, Branchu P, Ecoffey C, Wodey E. Douleurs chroniques après césarienne: Impact et facteurs de risque associés. Ann Fr Anesth Reanim. 2013;32: 772–778. doi:10.1016/j.annfar.2013.08.007

12. Moriyama K, Ohashi Y, Motoyasu A, Ando T, Moriyama K, Yorozu T. Intrathecal administration of morphine decreases persistent pain after cesarean section: A prospective observational study. PLoS One. 2016;11: 1–13. doi:10.1371/journal.pone.0155114

13. Brotons MJS, Echevarria M, Turmo M, Almeida C. Chronic Pain and Predictive Factors in the C-Section Surgery. J Anesth Clin Care. 2016;3: 1–6. doi:10.24966/acc-8879/100015

---

## [Editor Report · Decision Letter 1]

21 Aug 2020

The incidence of chronic pain following Cesarean section and associated risk factors: a cohort of women followed up for three months

PONE-D-20-14430R1

Dear Dr. Natalia Carvalho Borges

We’re pleased to inform you that your manuscript has been judged scientifically suitable for publication and will be formally accepted for publication once it meets all outstanding technical requirements.

Kind regards,

Ehab Farag, MD FRCA FASA

Academic Editor

PLOS ONE

---

## [Editor Report · Acceptance letter]

27 Aug 2020

PONE-D-20-14430R1 

The incidence of chronic pain following Cesarean section and associated risk factors: a cohort of women followed up for three months 

Dear Dr. Borges:

I'm pleased to inform you that your manuscript has been deemed suitable for publication in PLOS ONE. Congratulations! Your manuscript is now with our production department. 

Kind regards, 

on behalf of

Dr. Ehab Farag 

Academic Editor

PLOS ONE